# Evaluation of non-invasive continuous physiological monitoring devices for neonates in Nairobi, Kenya: a research protocol

Amy Sarah Ginsburg [ID],[1] Evangelyn Nkwopara,[2] William Macharia,[3] Roseline Ochieng,[3] Mary Waiyego,[4] Guohai Zhou,[5] Roman Karasik,[6] Shuai Xu,[7,8] J Mark Ansermino[9]

## ABSTRACT

**Introduction** Continuous physiological monitoring devices are often not available for monitoring high-risk neonates in low-resource settings. Easy-to-use, non-invasive, multiparameter, continuous physiological monitoring devices could be instrumental in providing appropriate care and improving outcomes for high-risk neonates in these low-resource settings.

**Methods and analysis** The purpose of this prospective, observational, facility-based evaluation is to provide evidence to establish whether two existing non-invasive, multiparameter, continuous physiological monitoring devices developed by device developers, EarlySense and Sibel, can accurately and reliably measure vital signs in neonates (when compared with verified reference devices). We will also assess the feasibility, usability and acceptability of these devices for use in neonates in low-resource settings in Africa. Up to 500 neonates are enrolled in two phases: (1) a verification and accuracy evaluation phase at Aga Khan University—Nairobi and (2) a clinical feasibility evaluation phase at Pumwani Maternity Hospital in Nairobi, Kenya. Both quantitative and qualitative data are collected and analysed. Agreement between the investigational and reference devices is determined using a priori-defined accuracy thresholds.

**Ethics and dissemination** This trial was approved by the Aga Khan University Nairobi Research Ethics Committee and the Western Institutional Review Board. We plan to disseminate research results in peer-reviewed journals and international conferences.

**Trial registration number** NCT03920761.

For numbered affiliations see end of article.

**Correspondence to**
Dr Amy Sarah Ginsburg; messageforamy@gmail.com

## Strengths and limitations of this study

► This research consists of two phases, a verification and accuracy evaluation phase and a clinical feasibility evaluation phase, and includes evaluation of two novel, investigational, non-invasive, multiparameter, continuous physiological monitoring devices.
► Verification of the reference devices is undertaken prior to initiating the accuracy evaluation of the investigational devices to ensure the reference devices are robustly functional and to confirm their within subject repeatability and accuracy compare to standard clinical measurements for the relevant parameters of interest.
► Reliability information gathered from the reference devices is used to determine specific a priori Go/No Go criteria for each parameter of interest and each investigational device.
► As with all measurements, there is uncertainty inherent in the measurements from the reference devices.
► Inability to control for the characteristics and conditions of the participating neonates and to standardise the environment and context are both strengths and limitations to interpreting the results.

## INTRODUCTION

In 2017 globally, 47% of all deaths in children under 5 years of age occurred within the first 28 days of life, which translates to a neonatal mortality rate of 18 deaths per 1000 live births or 2.5 million newborn deaths.[1] Sub-Saharan Africa bears the greatest burden of neonatal mortality with an estimated 1 million newborn deaths in 2017. Further efforts, especially in African countries, are needed to push progress towards achieving the Sustainable Development Goal target of reducing global neonatal mortality to 12 deaths per 1000 live births by 2030.[2] Without accelerated improvements, it is projected that 1.8 million neonates will die in 2030.[3] Innovations in neonatal care, particularly technologies that allow for early detection and intervention for major morbidities, hold great promise in helping to reduce current and projected neonatal mortality rates.

Multiparameter continuous physiological monitoring devices could be instrumental in identifying neonates at risk. We can then direct care provided for a neonate through automatic interpretations of vital signs that

help identify critical events and determine if treatment is sufficient or insufficient, ultimately improving newborn outcomes.[4 5] These devices would be most useful in low-resource settings where the need for such technologies is greatest. While continuous physiological monitoring is standard of care in high-resource settings for those who require it, the devices are expensive and require specialised training to operate, making them unsuitable for application in low-resource settings. To address these barriers, it is necessary to explore how these technologies can be adapted and/or optimised for use in low-resource settings. Ideally, the devices should be low cost, operator-independent, non-invasive and highly efficient in diagnostic performance and operator workload. This requires development of a robust testing platform that appropriately mimics conditions common in African newborn or neonatal intensive care units that would allow these type of technologies to be evaluated for feasibility and performance.

The Evaluation of Technologies for Neonates in Africa (ETNA) project was conceived with the goal of advancing and supporting development, as well as evaluation, of select devices for use in neonates in low-resource settings. By establishing a testing platform in an African site, and working collaboratively with partners with expertise in device development and evaluation and neonatal and child health, the project seeks to boost development and optimisation of promising newborn care devices that could be applied in low-resource settings in Africa. We acknowledge the many challenges involved in implementing such devices in low-resource settings (eg, electricity and internet access, behavioural change communication, etc), and the need to consider these challenges carefully prior to introduction. The purpose of this initial research is to produce evidence regarding the performance of two existing non-invasive, multiparameter, continuous physiological monitoring devices developed by device developers, Early Sense and Sibel, to accurately and reliably measure vital signs in neonates (when compared with verified reference devices) and to assess the feasibility, usability and acceptability of these devices for use in neonates in a low-resource setting in Africa.

## METHODS AND ANALYSIS
### Study design and setting
The primary objectives of this prospective, observational, facility-based research are: (1) to assess agreement between repeat observations by the investigational device and the reference device for each relevant measurement parameter of interest based on a priori-determined accuracy threshold among neonates; (2) to compare clinical event detection performance between the investigational device and the reference device and (3) to determine whether the investigational device is feasible, usable and acceptable to hospital administrators, healthcare providers and caregivers of neonates. Secondary

objectives include: (1) assessing diagnostic performance for each relevant measurement parameter of interest based on sensitivity, specificity, positive predictive value and negative predictive value compared with the reference device; (2) determining the downtime performance of the investigational device; (3) determining the alarm rate (events/hour) and the number of true/false alarms of the investigational device compared with the reference device; (4) determining the delay time between the investigational device and the reference device in true events and (5) determining the number of adverse device effects (ADEs) and serious adverse events (SAEs) during the use of the investigational device.

Beginning in June 2019 and anticipated to last approximately 18 months in Nairobi, Kenya, this research consists of two phases: (1) a verification and accuracy evaluation phase conducted at Aga Khan University-Nairobi (AKU-N), a private, not-for-profit university teaching hospital with a neonatal intensive care and high dependency units and (2) a clinical feasibility evaluation phase conducted at Pumwani Maternity Hospital (PMH), the largest referral maternity hospital in sub-Saharan Africa with no neonatal intensive care or high dependency units.

### Study participants
Up to 500 neonates, corrected age of ≤28 days admitted for routine observation and care at AKU-N and PMH are recruited by trained study staff during routine intake and screening procedures. To avoid potential selection bias, neonates are screened for enrolment in a sequential manner, as much as possible. Trained study staff assess the neonate for all inclusion and exclusion criteria (table 1). Final eligibility determination is dependent on the results of the medical history, clinical examination, appropriate

| Table 1 | Eligibility criteria |
|---|---|
| **Eligibility criteria** | |
| Inclusion criteria | Male or female neonate, corrected age of <28 days |
| | Willingness and ability of neonate's caregiver to provide informed consent and to be available for follow-up for the planned duration of the study |
| Exclusion criteria | Receiving mechanical ventilation or continuous positive airway pressure |
| | Skin abnormalities in the nasopharynx and/or oropharynx |
| | Contraindication to application of skin sensors |
| | Known arrhythmia |
| | Any medical or psychosocial condition or circumstance that, in the opinion of the investigators, would interfere with the conduct of the study or for which study participation might jeopardise the neonate's health |

understanding of the study by the caregiver and completion of the written informed consent process. A neonate may be enrolled to the study more than once as long as they meet the eligibility criteria and the caregiver(s) is willing to have the neonate participate.

For the feasibility, usability and acceptability assessment, hospital administrators and study healthcare providers are enrolled if they are 18 years or older, involved in or aware of the ETNA study, and have provided written informed consent. Caregivers may be enrolled if they are 18 years or older, have a neonate enrolled in the study and are willing to participate in an in-depth interview as well as direct observation while their neonate is on or attached to the investigational device(s).

### Investigational devices
Developed by Israeli-based EarlySense since 2009, the Insight system, released in 2016, is a contact-free monitoring system composed of a small piezoelectric sensor pad that can be placed under the patient's mattress, and is designed to measure and record a patient's heart rate (HR), respiratory rate (RR), motion and sleep status.[6] Information from the sensor pad, in combination with Early Sense's artificial intelligence analytics, is transmitted to a monitor to provide alert indications and vital sign trends to healthcare providers so that they can monitor changes in a patient's condition. Currently in use in hospitals, rehabilitative centres, and nursing homes to measure vital signs in adults and children above 10 kg, the device is modified for use in neonates as part of this study. The adult device received regulatory approval from the US Federal Drug Administration and has a Conformité Européene mark for continuous and contactless measurement of HR, RR and motion. No adverse events (AEs) related to the system have been reported during 10 years of monitoring.

Developed in 2019, the advanced neonatal epidermal (ANNE) system from Sibel, a technology company spun out from the Center of Bio-Integrated Electronics at Northwestern University in the USA, is a system of two time-linked soft and flexible sensors designed to measure and monitor vital signs including HR, RR, oxygen saturation ($SpO_2$) and skin temperature in neonates.[7] The chest sensor couples to the skin via a hypoallergenic, biocompatible hydrogel adhesive optimised for reduced peel force on removal, and the limb unit couples via a latex-free soft fabric wrap adaptable to a range of foot sizes and anatomies. Information from the sensors is wirelessly transmitted to a monitor or mobile device via encrypted Bluetooth for real-time streaming from a customised mobile software application as well as onboard memory storage on the sensors themselves. The device has been validated in more than 50 neonates in a neonatal care unit without AEs.

### Reference devices
We are employing the Masimo Rad-97 and the Spengler Tempo Easy Bleu devices as our reference devices for this study. The Masimo Rad-97 provides continuous physiological monitoring of HR, RR, $SpO_2$ and capnography. The Spengler Tempo Easy Bleu non-contact infrared thermometer predicts core body temperature from the temporal artery temperature.

### Study procedures
Following completion of screening for eligibility, a study comprehension checklist and written informed consent, study staff perform procedures (online supplementary appendix 1: Schedule of study procedures and evaluations) according to the most recently approved version of the protocol (current V.1.1, 18 June 2019). Enrolled neonates are assigned a participant identification number; information is collected on sociodemographic characteristics, current clinical status, medical history, medications; and a physical examination is performed.

Prior to initiating the accuracy evaluation of each investigational device, verification of the reference devices, Masimo Rad-97 and Tempo Easy Bleu, is undertaken at AKU-N to ensure they are robustly functional and to confirm their within subject repeatability and accuracy compared with standard clinical measurements (eg, manual, bedside electrocardiography) for the relevant parameters of interest. Neonates enrolled during reference device verification continue to receive local standard of care while being observed intermittently for vital signs collection for a minimum of 1 hour using the Masimo Rad-97 and intermittent measurements with the Tempo Easy Bleu. Observations may include video recordings of the neonate and the Masimo Rad-97 reference device monitor for later review to facilitate manual count observations. The reference device measurements will be compared with manual measurements, clinical monitor observations and video-assisted observations. Reliability information gathered from the reference devices is used to determine specific Go/No Go criteria for each parameter and each investigational device. Further evaluation of each investigational device only proceeds should these criteria be met.

Enrolment in the accuracy evaluation of the investigational devices, EarlySense Insight system and Sibel ANNE system, is initiated at AKU-N to formally assess their accuracy compared with the verified reference device using repeated observations. Enrolled neonates continue to receive local standard of care while having vital signs collected from the reference device as well as one or both of the investigational devices. Placement of the investigational and reference devices is done in a manner so as not to interfere with the neonate's clinical care. Observations are collected for a minimum of 1 hour and potentially for the entire duration of their stay in the hospital. Observations may consist of videotaping and/or taking photos of the neonate during the observation period after obtaining informed consent from the caregiver. Any photos or videos takes are identified by patient identification number only and stored on a secure server until the analyses are completed and destroyed following analyses.

During the observation, clinical status and any activities are updated and recorded including type and duration of care activities (eg, feeding, diaper changes, bathing, kangaroo mother care, etc), clinical procedures, interventions, therapies, laboratory tests, medications, environmental features and exposures during the hospitalisation. The device placement, output and signal quality are also monitored. In addition, the neonates are assessed for any safety issues. Agreement between the investigational and reference devices is determined using a priori-defined accuracy thresholds. Thresholds are determined largely based on repeated within and between subject observations during verification of the reference devices. This is complemented by previously published international standards where available, and clinical expert consensus opinion as needed. Two a priori-determined thresholds are determined: one lower threshold to allow the device developer to optimise the device for retesting, and a second higher threshold to allow the device to move on to the clinical feasibility phase of testing. A maximum of five rounds of testing and retesting are permitted for each investigational device. Each round of testing or retesting consists of using a cohort of 20 neonates. Should the lower threshold not be reached for at least one parameter, no further testing of the investigational device is performed. Thus, information collected during the accuracy evaluation along with the a priori-determined Go/No Go criteria established during verification of the reference devices define which, if any, of the investigational devices moves forward with additional rounds of testing or into the clinical feasibility evaluation phase at PMH.

An investigational device advances to the clinical feasibility evaluation phase once the agreement for the measurement parameters of interest exceed the higher accuracy threshold. Enrolment in the clinical feasibility evaluation phase of the investigational devices occurs at PMH up to 120 enrolled neonates who receive local standard of care while being monitored with the reference device(s) and one or both of the investigational devices. Observations are collected for a minimum of 1 hour and involve measurement of vital signs via the investigational and reference devices and monitoring for any critical event (ie, low or high HR, RR or temperature or oxygen desaturation and apnoea). Agreement between repeated observations from the investigation a land reference devices as well as diagnostic performance in clinical event detection is evaluated. Additional performance metrics such as alarm rates, alarm delays and uptime\downtime are compared between the investigational and reference devices. Participation in the study does not interfere with or unnecessarily delay the clinical care of the neonates.

Throughout all phases of the research, the investigational devices are not used to inform clinical care. During the clinical feasibility evaluation phase, ETNA site study staff and hospital healthcare providers are blinded to the data collected from the investigational devices to prevent interference with clinical care. The study site investigators are responsible for close safety monitoring

of all participating neonates, including assessing for and reporting ADEs (eg, erythema or oedema at the investigational or reference device sensor site) and/or SAEs (ie, any ADE resulting in permanent skin damage). Any ADEs or SAEs will be treated until resolution or stabilisation, and may require removal of devices and withdrawal of the neonate from the study if necessary. If withdrawn by the study team, any enrolled neonate who completes at least 1 hour of monitoring will be included in the analysis and results.

### Qualitative substudy

After written informed consent is received from the study participants, a mixed-methods evaluation and data collection through audio-recorded semistructured in-depth interviews and direct observations are conducted by trained qualitative study staff to assess the feasibility, usability and acceptability of the investigational devices for monitoring of neonates in an African setting. Questions around technology use, experience with continuous monitoring devices and specific to each investigational and reference device will be asked and their use observed. All hospital administrators and study healthcare providers may be involved in this portion of the study. Caregivers with a neonate enrolled in the study may also be asked if they would like to participate in the qualitative portion of the study.

### Sample size

A total of up to 500 neonates are enrolled. For the verification of the reference devices at AKU-N, up to 30 neonates are enrolled. Once this initial testing and data collection of the reference devices are complete, for the accuracy evaluation phase at AKU-N, up to 120 neonates per investigational device are enrolled. Sample size estimates for the verification of the reference devices and the accuracy evaluation phase are based on the CIs desired for the limits of agreement. Sample sizes of 100–200 typically provide tight CIs. A sample of 20 neonates with 10 replications per neonate per device per round of testing provides limits of agreement with 95% CIs±0.24, calculated as $1.96*sqrt(3/(20*10))$, times the SD of the paired differences. The paired differences are from the reference device and manual measurements obtained during verification of the reference device, and from the reference device and investigational device measurements obtained during the accuracy evaluation phase. For the clinical feasibility evaluation phase at PMH, up to 120 neonates per investigational device are enrolled. The sample sizes for each phase have been selected to maximise the amount of information collected within the confines of the available resources.

For the feasibility, usability and acceptability assessment, the total sample size includes all hospital administrators and study healthcare providers willing to participate and provide consent as well as up to 30 caregivers willing to participate and provide consent study at each site.

## Data collection and quality assurance

Quantitative study data are collected by clinical study staff using designated source documents as well as electronic or paper-based case report forms. Data are stored and managed by a database developed via Research Electronic Data Capture, a secure web application. Continuous physiological data and event data are recorded from the investigational and reference devices at least once a second. All electronic data are collected wirelessly or via a wired connection from the investigational and reference devices to a study laptop using custom software applications. Qualitative study data are collected using paper-based forms and audio recordings, which are subsequently transcribed for analysis.

Clinical research data, including data collected from the investigational and reference devices, are maintained through a combination of secure electronic data management system and physical files with restricted access to ensure confidentiality. Two distinct study databases are maintained separately: the primary study database and a database with participating neonate's personally identifiable information. To ensure accuracy and completeness, data are routinely reviewed by the investigators through quality assurance reviews, audits and evaluation of the study safety and progress. Guideline for Good Clinical Practice (GCP)/ISO 14155 compliance is followed to ensure accurate, reliable and consistent data collection.

## Data management

Primary data management activities, which include deidentified investigational and reference device data transfer using end-to-end encryption with two-factor authentication, data entry and validation, data cleaning, database quality control and disaster recovery plans are undertaken at the study site and are overseen by the on-site data manager. Data review and analysis, oversight and preparation of final study database is performed by the investigators in collaboration with the study site. Data are maintained and stored securely in databases hosted at the study site throughout the study and for at least 5 years after study closure. All data management activities are in compliance with International Council on Harmonisation (ICH) GCP E6, sponsor organisation and institutional requirements for the protection of children and confidentiality of personal and health information.

## Outcomes

We hypothesise that the investigational device is accurate and reliable compared with the reference device for each relevant measurement parameter of interest among neonates and is feasible, usable and acceptable for use in neonates in low-resource settings. The primary endpoint and secondary endpoints are detailed in box 1.

## Statistical analyses

Every second of data is automatically graded as optimal, acceptable and unacceptable based on predefined rules for each device and each measurement parameter of

---

**Box 1  Study endpoints**

Primary endpoints
► Agreement of the relevant measurement parameters of interest between the investigational device and the reference device at each observation.
► Agreement of clinical event detection between the investigational device and the reference device at each observation.
► Feasibility, usability and acceptability of the investigational device among hospital administrators and healthcare providers.
► Acceptability of the investigational device among caregivers.

Secondary endpoints
► Diagnostic performance of the investigational device to appropriately identify the following critical events:
  – Low heart rate.
  – High heart rate.
  – Low respiratory rate.
  – High respiratory rate.
  – Oxygen desaturation.
  – Apnoea.
  – Low temperature.
  – High temperature.
► Downtime duration of the investigational device.
► Alarm rate (events/hour and ratio of false positives to missed critical events of the investigational device's alarms compared with the reference device's alarms.
► Response time of the investigational device's alarms compared with the reference device's alarms for critical events
► Proportion of neonates with adverse device effects and serious adverse events resulting in skin damage.

---

interest according to the quality of the data for each measurement parameter of interest. The Masimo Rad-97 provides a signal quality index that is used to determine data quality for HR and $SpO_2$. A custom algorithm has been produced to determine the capnography signal quality index. Each of the investigational devices also provides a signal quality index. The quality thresholds are determined following verification of the reference devices. All comparisons are performed from observations between two devices (or a single device during verification). At least 10 observations of 60 s of optimal quality data in each neonate, at least 5 min apart, are randomly selected for each measurement parameter of interest from the full recording. For the clinical feasibility evaluation phase, accuracy comparisons use optimal or acceptable data. At least 3 hours of recording to a maximum of 12 hours are used for the performance metrics such as alarm rates, alarm delays and uptime\downtime.

The repeatability of the reference device parameter estimates initially is assessed with the intraclass correlation coefficient (ICC). Additional training or standardisation of procedures is performed to ensure at least good repeatability (ICC >0.7). This is followed by measuring agreement between the repeated reference observations and between the manual, clinical monitor and video-assisted methods and the reference observations using the methods described by Bland and Altman for replicated observations.[8] The agreement is reported as a mean

bias with 95% CIs and 95% limits of agreement. Graphical representation of the data is assessed with agreement plots, Clarke error grids and Polar plots to identify extreme outliers and significant data trends.

In the accuracy evaluation, the root mean square difference and ICC are calculated for each measurement parameter of interest to compare the multiple repeated observations between the investigational and reference devices. The agreement between each investigational device and reference device(s) is then calculated using the methods described by Bland and Altman for replicated observations. The agreement is reported as a mean bias with 95% CIs and 95% limits of agreement. Graphical representation of the data is be assessed with agreement plots, Clarke error grids and Polar plots to identify extreme outliers, impact on clinical decisions, and significant data trends. An a priori-defined accuracy margin for agreement is used as a threshold value to allow for decisions regarding proceeding to additional testing.

In the clinical feasibility evaluation phase, agreement between each investigational device and reference device(s) is assessed as in the accuracy evaluation phase. Event detection rates, alarm rates, alarm delays and uptime/downtime are summarised with means, medians SD and IQRs as appropriate. Summaries of sensitivity, specificity, positive predictive values and negative predictive values comparing each measurement parameter of interest in the investigational device(s) to the reference device(s) are produced. Comparisons of binary events are assessed using Cohen's weighted Kappa and McNemar's test. The non-inferiority of alarm rates, alarm delays and uptime/downtime are evaluated based on prespecified thresholds.

Qualitative data are collected through in-depth interviews and/or semistructured questionnaires and analysed to assess feasibility, usability and acceptability of the investigational devices among hospital administrators and healthcare providers, and acceptability among caregivers of enrolled neonates. Questions that explore familiarity, knowledge, perceptions, attitudes and behaviours regarding the devices are included. The qualitative data are in narrative format and the results are descriptive. The questionnaires are coded and analysed using a codebook with identified themes, including feasibility of using each investigational device, barriers and facilitators to use, and perceived value. Qualitative data analysis software is used to organise, code and analyse the qualitative data in an iterative process. The study team starts by identifying an initial set of codes and themes based on the categories from the interview guides. During the coding process, attention is paid to identifying emergent issues and themes that are added to the codebook and included in the analysis. Responses from the interviews are coded and discrepancies discussed and resolved for the final analysis and theme identification.

## Ethics and dissemination
### Ethical approvals and consent
The study is conducted in accordance with the ICHGCP and the Declaration of Helsinki 2008. The protocol and other relevant study documents study were approved by the Western Institutional Review Board 20 191 102 (Puyallup, Washington, USA), and the Aga Khan University Nairobi Research Ethics Committee 2019/REC-02 (v2) (Nairobi, Kenya). Written informed consent is obtained in the local language by trained study staff from all eligible neonate's caregivers and for the qualitative substudy, from participating hospital administrators, healthcare providers and caregivers prior to enrolment. Potential participants will have adequate time to ask questions and a comprehension checklist will be administered to ensure participant understanding.

### Possible risks
Caregivers may feel compelled to enrol in the study in order to receive care for their neonate within a research setting, which may be perceived as of a higher quality than the standard of care. In order to minimise the risk of coercion, during the informed consent process, study staff emphasise that the neonate will receive the required medical care whether enrolled in the study or not. Other potential risks to study participation may include those associated with the placement and attachment of the investigational and reference devices, and delayed medical management. Study staff are trained in the appropriate placement of investigational and reference devices' sensors to minimise discomfort to the neonates as well as to avoid interference with any assessment, treatment or intervention necessary for clinical care. There is a potential risk of skin irritation with the ANNE sensor system and neonates will be closely monitored and treated for any AEs. Study staff are also trained in integrating study procedures with clinical care and to always prioritise clinical care above study procedures. Extreme care is taken to ensure that no necessary treatment is delayed to accommodate study procedures.

### Dissemination
We plan to disseminate study results in peer-reviewed journals and international conferences, targeting those involved in the clinical care of neonates in low-resource settings as well as those who develop and advise on policies and guidelines in those settings.

### Efforts towards rigorous protocol
Dedicated study staff trained in GCP, operation, use and maintenance of the investigational and reference devices, and study-specific procedures follow neonates enrolled in the trial to assure the protocol and standard operating procedures are followed and data are accurately collected. Standardised study-specific training, supervision and oversight are undertaken to ensure quality, consistency and harmonised trial procedures and implementation. Regular monitoring is provided by the coinvestigators

to assess compliance with human subjects and other research regulations and guidelines, adherence to the study protocol and procedures, and quality and accuracy of data collected.

## Limitations and bias

Limitations to this study and potential sources of bias include the sampling strategy, the uncertainty inherent in the measurements from the reference devices, the limited standardisation of time of day of recording and the inability to control the conditions and standardise the context. Because there is a large variation in the various ages, weights, sizes, disease states, clinical presentations, interventions received and conditions of the participating neonates, it is not possible to control for all these variables. Likewise, the environment cannot be controlled, does not allow for complete standardisation and may introduce additional sources of bias. These limitations may also be viewed as strengths.

**Author affiliations**
[1]Clinical Trial Center, University of Washington, Seattle, Washington, United States
[2]Children's Healthcare of Atlanta, Atlanta, Georgia, United States
[3]Pediatrics, Aga Khan University, Nairobi, Kenya
[4]Pediatrics, Pumwani Maternity Hospital, Nairobi, Kenya
[5]Biostatistics, Brigham and Women's Hospital, Boston, Massachusetts, United States
[6]EarlySense, Ramat Gan, Israel
[7]Sibel Inc, Evanston, Illinois, United States
[8]Dermatology, Northwestern University, Evanston, Illinois, USA
[9]Anesthesiology, The University of British Columbia, Vancouver, British Columbia, Canada

**Acknowledgements** We would like to thank Dustin Dunsmuir who wrote the IAP logger application used to collect the high resolution data from the reference device.

**Contributors** ASG, EN, WM and JMA designed the study and wrote the protocol. RO, MW, GZ, RK and SX reviewed and provided critical input to the study design and protocol. ASG wrote the first draft of the manuscript, and EN and JMA provided additional input. All authors worked collaboratively, reviewed the manuscript and made the decision to submit the final manuscript for publication.

**Funding** This work is supported by grants from the Bill & Melinda Gates Foundation (OPP1203136) and the Save the Children Innovation Council. The authors had final responsibility for the decision to submit this manuscript for publication.

**Competing interests** RK is employed by EarlySense and SX is employed by Sibel.

**Patient and public involvement** Patients and/or the public were not involved in the design, or conduct, or reporting, or dissemination plans of this research.

**Patient consent for publication** Not required.

**Provenance and peer review** Not commissioned; externally peer reviewed.

**ORCID iD**
Amy Sarah Ginsburg http://orcid.org/0000-0002-2291-2276

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
