## [Reviewer comments · BMJ Open]

ARTICLE DETAILS

TITLE (PROVISIONAL)	Evaluation of noninvasive continuous physiological monitoring devices for neonates in Nairobi, Kenya: A research protocol
AUTHORS	ginsburg, amy sarah; Nkwopara, Evangelyn; Macharia, William; Ochieng, Roseline; Waiyego, Mary; Zhou, Guohai; Karasik, Roman; Xu, Shuai; Ansermino, J. Mark

VERSION 1 - REVIEW

REVIEWER	Ayesha Johnson Florida Department of Health USA
REVIEW RETURNED	04-Nov-2019

GENERAL COMMENTS	1. I am still not totally clear on why the investigational device is being compared to the reference device for eventual use in low resource setting. Is it that the reference device is too costly to use in a low resource setting? Is the study comparing the investigational device to the reference device in a low resource setting? Or a high resource setting? If it is being compared in a high resource setting, how will this inform its use in a low resource setting? 2) There were a few grammatical errors. The reviewer provided a marked copy with additional comments. Please contact the publisher for full details.
--

REVIEWER	Professor Celia Harding City, University of London, UK.
REVIEW RETURNED	07-Nov-2019

GENERAL COMMENTS	Dear Authors, Thank you for submitting this important study protocol. Given the environment where you will be completing your intervention, this is a vital piece of work. I have a few amendments to suggest. I have recommended a further evaluation of statistical methods, not because I can see anything unusual in your methods, but to ensure that your protocol is robust. Another person looking specifically at this are would be highly beneficial. I note your funding source, etc. However, I am unclear as to use of any standard checklists such as CONSORT, STROBE, PRISMA, etc., and this may be useful just for clarity's sake.
--

	You have relevant information on your participant information sheets and consent forms in your appendices which should also be in the body of your text, e.g. you need to add how long you will give potential participants to read the information, and then sign the consent form; how you will address potential literacy problems within your participating group - in terms of reducing risk of coercion; how you will ensure overall that coercion will not occur; that you will not penalise anyone if they elect to drop out of the study - and that participants can drop out at any time. Do you have any clear exclusion criteria? If so, what are they? Will you be including all infants , regardless of individual problems? I have noted your statement re: withdrawal of infants from the study as appropriate, but it may be useful to consider and state the impact of including infants with possible neurodisability within the body of your sample. Of course, there may be no significant issues, but I think you should mention that any participants who will not complete taking part will be clearly stated in your findings. Please can you state how long data will be stored, where it will be stored, and how it will be destroyed at the end of the storage period? You need to be clear about dates in your proposal. How long will the study last? Your current protocol is dated June 2019 - can you please clarify if this is the enrolment date? What will your anticipated start date be? Please can you change use of [HCPs] for "Healthcare Providers"? This term typically refers to [Health Care Professions] in the literature, and therefore maybe misleading. I suggest "HPs" instead. Page 13, line 212: One of your observations will be during feeding. Will this be tube feeding, oral feeding, i.e. breast / bottle? It is important to differentiate as there are likely to be differences in instrumental readings between different methods, and also if this is one of the first oral feedings offered. Will you be using any early feeding / infant state checklists as part of your observations? Thank - you.
--	--

REVIEWER	Dr Michelle Scoullar Burnet Institute Australia
REVIEW RETURNED	20-Nov-2019

GENERAL COMMENTS	Manuscript Number: BMJOpen 2019-035184 Full Title: Evaluation of noninvasive continuous physiological monitoring devices for neonates in Nairobi, Kenya: A research protocol Article Type: Protocol Authors: Amy Sarah Ginsburg,* Evangelyn Nkwopara, William Macharia, Roseline Ochieng, Mary Waiyego, Guohai Zhou, Roman Karasik, Shuai Xu, and J. Mark Ansermino Summary:  • Overall this is a clearly presented and thoroughly thought through protocol on an important area of clinical care and in a particular area of need, that is increasing access to and usability of equipment for clinical care in resource constrained settings. It is clearly important that research that continues to expand and enable a deeper capacity to provide quality care for sick newborns be encouraged and I commend the authors on this intent and work.. • My main comment is that the manuscript would benefit from
--

a section that highlights the study team is well aware of the many challenges that occur when implementing equipment in certain settings and what the study team intends to do to address these were the investigational devices included in this study prove useful and therefore potentially be considered for broader use in more remote settings. Not only the infrastructural challenges (e.g. electricity in remote settings etc), but particularly the human factors that make implementing new processes / devices / treatment options challenging in any clinical setting, and how the study team intends to address the factors that contribute to this.

Minor comments:

- Line 134 I assume this is meant to be less than or equal to 28 days not more than. Again in table 1 you have used the symbol for older than 28 days.
- Line 150 ?since 2009
- Regarding the EarlySense device:
 - o Given it is already used in adults and children > 10kgs, it would be good to know what some of the evaluations of this device have shown in terms of accuracy and feasibility, particularly if there have been any important learning points regarding implementation in a busy hospital setting.
 - o Line 162 is there a reference relevant to the reporting of adverse events? Has there been a formal independent evaluation of this or is it the company's evaluation of adverse events?
 - o What is the output from the motion and sleep status monitoring? E.g. binary asleep / awake; moving / not moving. Or is there more nuance? What is the purpose of this monitoring? Would it detect potential seizure activity or is it to detect inactivity / trying to warn of potential Brief Resolved Unexplained Events (BRUEs) or possible SID events.
- Regarding both devices:
 - o What environmental conditions can these devices operate in? specifically temperature and humidity limits of operation? What fluctuations can they tolerate in these parameters, e.g for shipping and storage of equipment.
- Reference devices:
 - o Why were these devices chosen? Could you provide details on their validation and accuracy including false alarm rates.
 - o What are the limitations of these reference devices in a tertiary setting and what are the specific limitations that make them inappropriate for use in a scaled-up fashion at smaller nurseries.
- Study procedures:
 - o What level of training will the study staff have, particularly in terms of examining the enrolled neonate and their ability to detect and act appropriately if any unexpected findings are identified on examination?
 - o Line 191: could you please detail what you mean by "standard clinical measurements." Manual measurement? What will this mean of O2 saturations?
 - o How will the videotaping be used? How long will these videos be stored for and where? How will the study team ensure there is no inappropriate use of these videos?
 - o What measures will the study team take to ensure any unintended interruptions to routine care are minimised? E.g will the study team detect if KMC is given less frequently to study babies through potential HCP concern re interrupting the study / measurements of the new device.
- Qualitative sub-study
 - o What number of hospital administrators and study HCPs do you anticipate collecting qualitative data from? Is it necessary to

	interview all? Are study HCPs different from hospital employed HCPs or the same? If the HCPs are invited to participate, what measures will the study take to try minimise bias secondary to HCPs self-selecting to participate?  • Outcomes:  o The two devices included in the study measure some variables that are not measured in the reference devices. How will these parameters be assessed for accuracy? o Is the intention that both devices would be used on each neonate? If so how would discrepancies between devices be interpreted? • Possible risks  o Could the study team please outline in a little more detail how / what measures will be taken to ensure no necessary treatment is delayed to accommodate study procedures • Inclusion / Exclusion criteria  o Why is known arrhythmia excluded? Wouldn't it be useful to know if this is accurately detected? • Primary endpoints  o Why can the 'relevant measurement parameters of interest' not be stated clearly so we know what these parameters are? • Full protocol  o There is some very interesting detail of the two facilities in the full protocol that is not in the manuscript. I think It would be interesting to readers to include a little more detail of the two study facilities in the manuscript so as to gain a better understanding of the extreme workload HCPs in these facilities face and how such devices could potentially be of huge benefit. o Sample size estimation:  □ Could you clarify why 25% difference in incidence rate was chosen? It seems quite a large difference to use as the cut off.
--	---

REVIEWER	Dr. S. Lava Pediatrics, University Hospital of Lausanne, Lausanne, Switzerland
REVIEW RETURNED	24-Nov-2019

GENERAL COMMENTS	Ginsburg and colleagues aim at investigating noninvasive multiparameter, continuous physiological monitoring devices in Kenya. Their rationale starts from the recognition of the high neonatal mortality in sub-Saharan Africa. They believe that innovations in technologies that allow for early detection of major morbidities “hold great promise in helping to reduce current and projected neonatal mortality rate”. Their study rationale therefore moves from the belief that “multiparameter continuous physiological monitoring devices could be instrumental in identifying neonates at risk” and that the two noninvasive continuous physiological monitoring devices might better perform, with respect to this task, than currently available, traditional monitoring devices, since, in their opinion, current “devices are expensive and require specialized training to operate”. The problem of high perinatal and neonatal mortality is a major worldwide problem, mainly driven by high mortality in low-income countries, like sub-Saharan Africa. Efforts to improve this situation are praiseworthy. Since a lot of such interventions are more political than medical, medical efforts should mainly aim at improving care, education and research. This study is therefore interesting.
--

However, I see a lot of problems with the rationale of this study:

- 1) Authors claim that "Innovations in neonatal care, particularly technologies that allow for early detection and intervention for major morbidities, hold great promise in helping to reduce current and projected neonatal mortality rates". Even if this is probably true, if they want to focus on sub-Saharan Africa, several other interventions should be prioritized and hold promise of much more impact: improving hygiene, delivery at the hospital or appropriate institutions, availability of trained personnel, preterm birth prevention, train personnel to recognize clinical signs of deterioration (e.g. hypoglycemia, sepsis, respiratory distress, ...), accessibility to clean water and basic medications, ... In fact, clinical suspicion has been demonstrated to be superior to any laboratory or apparative diagnostic tool in detecting risky clinical situations.
- 2) Authors claim that "continuous physiological monitoring is standard of care in high-resource settings" but this only apply to about 10% of newborns, who are born preterm or who need some specific care. >80% of newborns do not need any monitoring, even in high-resource countries.
- 3) It is unclear, while these devices should directly be tested in Africa. The potential of user-friendly manipulation and of cheaper costs also applies to high-income countries. Performing research with the fragile African population (who might be induced to participate in research by the hope of receiving better care) needs a solid rationale (either devices explicitly developed and designed for the use solely, or at least mainly, in Africa, or second-step research after having been studied in high-income countries). Were these devices developed explicitly for use in Africa? Was it verified that these populations will (be able to) buy these devices? Will they be so cheap?
- 4) Are these devices really cheaper than traditional monitors? Please provide a direct cost comparison.
- 5) "This requires development of a robust testing platform that appropriately mimics conditions common in African newborns" (lines 93-94). Sorry, but I do not agree. You might also directly test the devices in the real-world setting of everyday clinical life, without the need of complicating the research process by developing a "robust testing platform". Furthermore, how is a robust platform defined? What does it need in order to be robust? Additionally, how it will be verified and tested that the platform is robust?
- 6) Lines 122-123: In Africa the problem is probably mainly not to detect the alarms, but to have enough trained people to appropriately react to the alarms and the availability of the consequent therapies (e.g. non-invasive and invasive ventilation, oxygen, antibiotics and drugs, nutritional supplements, ...). Even a step before, the main problem is probably availability of health-care institutions and hospitals (with adequate electrical energy and oxygen supplies). To alarm people without the possibility of appropriately reacting to the detected situations is of no interest.
- 7) Newborns can be enrolled several times (lines 140-141), which might insert several biases.
- 8) Lines 217-219: thresholds should be clearly defined in the protocol, and not be a work-in progress process. According to the current manuscript and protocol, thresholds will be determined by an arbitrary combination of repeated observations performed on the reference device during the verification phase, international standards, and clinical expert consensus opinion. This definition is not enough clear and is therefore not really "a priori defined".
- 9) Your study design is "dangerous" with respect to study feasibility and solidity. In your first step, you evaluate just 2 devices. If they will

not pass the first phase, your study will already stop. If you want to keep such a “2-step design”, you should include in this first phase much more devices (for example, with the aim of finally selecting the 2 best performing devices). In this version, your study entails a big risk to be abandoned already after the very few steps (but after having already included several children).

Several further points are also critical.

- Arbitrary choice of the sample size, without any calculation
- The manuscript is much too long.
- This manuscript more resembles a submission to the Ethical committee than a scientific paper: several “standard protocol sentences”, which are however general and not specific, or even do not add any relevant content to the study methods (e.g. “Standardized training, supervision, and oversight are undertaken to ensure quality, consistency, and harmonized trial procedures and implementation”: this is generally requested for every trial, but... what does the reader know, after reading this sentence, which he did not know before? Such a sentence is mandatory in a study protocol for ethical submission (and is often copy-pasted from study protocol templates), but does not add anything to a paper presenting the study methods, with the aim of being shared with the medical and scientific community...).
- Line 121: what do you mean with “downtime performance”? In a downtime period, a device is supposed not to work... how can its performance during a “break” time be evaluated? Why should this be relevant?
- Line 124: “delay time”: is the device so slow in transmitting information to have a measurable delay? Should the transmission not be instantaneous (the delay between clinical event, detection by the device and transmission to the central being <1s)?
- Lines 127-128: what is the difference between “verification” and “accuracy evaluation”? Please be as explicit and detailed as possible.
- Line 254: “qualitative”: you mean “quantitative”, well?
- Line 256: “observations [...] by trained qualitative staff”: how will they be trained? What will they assess and how? This description is too “general” and not enough detailed and objective. Please provide a list of items/data they will address, observe and collect.
- Lines 335-336: “An a priori–defined accuracy margin for agreement is used”: please transparently state this a priori defined margin (with its minimum and maximum) in the protocol.
- Lines 341-342: “sensitivity, specificity, positive predictive values and negative predictive values”: for which events? How will these events be defined and decided? (e.g. a possible event might be “desaturation”. This should be defined as “desaturation at <...%”, with rationale for the choice of that specific value).
- Lines 347-348: which questions will be asked in these semi-structured questionnaires? In a paper, which uses 20 pages to describe the study methods, this information must be provided.
- Line 365: why approval (also) from USA, if the study is performed in Kenya?
- Line 381: “ANNE sensor system and neonates will be closely”  “ANNE sensor system, if this device will pass the validation phase, and neonates will be closely”
- Lines 401-403: are these really limitations? I feel, they are rather strengths: a good device should be reliable in a wide range of “timepoints of day”, clinical conditions, gestational age, weight range, stability degrees, ...
- Line 407: again, this is not a limitation. You are not evaluating a

	therapy, but the reliability of a diagnostic device: it should be reliable across a large spectrum of situations. If the test is standardized, the results are not generalizable and, therefore, not extrapolable to everyday clinical life.
--	--

VERSION 1 – AUTHOR RESPONSE

Reviewer(s)' Comments to Author:

Reviewer: 1

Reviewer Name: Ayesha Johnson

Institution and Country: Florida Department of Health

USA

Please state any competing interests or state 'None declared': None declared

Please leave your comments for the authors below

1. I am still not totally clear on why the investigational device is being compared to the reference device for eventual use in low resource setting. Is it that the reference device is too costly to use in a low resource setting? Is the study is comparing the investigational device to the reference device in a low resource setting? Or a high resource setting? If it is being compared in a high resource setting, how will this inform its use in a low resource setting?

Response: The investigational devices are being compared to the reference devices in a low-resource setting because the goal is to further develop and evaluate these investigational devices for use in low-resource settings. As we had noted in the Introduction section, "The purpose of this initial research is to produce evidence regarding the performance of two existing noninvasive, multiparameter, continuous physiological monitoring devices developed by device developers, EarlySense and Sibel. The intent is to provide evidence to establish whether these investigational devices can accurately and reliably measure vital signs in neonates (when compared to verified reference devices) and to assess the feasibility, usability and acceptability of these devices for use in neonates in a low-resource settings in Africa."

2) There were a few grammatical errors.

Response: Noted. We have reviewed for grammatical errors and tried to make corrections.

Reviewer: 2

Reviewer Name: Professor Celia Harding

Institution and Country: City, University of London, UK.

Please state any competing interests or state 'None declared': None declared.

Please leave your comments for the authors below

Dear Authors,

Thank you for submitting this important study protocol. Given the environment where you will be completing your intervention, this is a vital piece of work. I have a few amendments to suggest.

I have recommended a further evaluation of statistical methods, not because I can see anything unusual in your methods, but to ensure that your protocol is robust. Another person looking specifically at this area would be highly beneficial.

Response: We defer to the editor for further review of the study's statistical methods.

I note your funding source, etc. However, I am unclear as to use of any standard checklists such as CONSORT, STROBE, PRISMA, etc., and this may be useful just for clarity's sake.

Response: Given this research is not a randomized controlled trial, observational study, nor a systematic review/meta-analysis, CONSORT, STROBE, or PRISMA reporting guidelines do not apply easily. STARD reporting guidelines are used for diagnostic accuracy studies (and we have reviewed them), but in this research, we are not diagnosing a condition or disease, but rather are just measuring continuous physiological variables, and comparing investigational device measurements to reference device measurements. None of the existing reporting guidelines apply very well. All testing is done in compliance with ISO 14155: Clinical investigation of medical devices for human subjects —

Good clinical practice (second edition 2011-02-01).

You have relevant information on your participant information sheets and consent forms in your appendices which should also be in the body of your text, e.g. you need to add how long you will give potential participants to read the information, and then sign the consent form; how you will address potential literacy problems within your participating group - in terms of reducing risk of coercion; how you will ensure overall that coercion will not occur; that you will not penalise anyone if they elect to drop out of the study - and that participants can drop out at any time.

Response: As requested, we have included more details regarding informed consent procedures in the Ethical approvals and consent section. Of note, in the Possible risks section, we had included, "In order to minimize the risk of coercion, during the informed consent process, study staff emphasize that the neonate will receive the required medical care whether enrolled in the study or not."

Do you have any clear exclusion criteria? If so, what are they? Will you be including all infants, regardless of individual problems? I have noted your statement re: withdrawal of infants from the study as appropriate, but it may be useful to consider and state the impact of including infants with possible neurodisability within the body of your sample. Of course, there may be no significant issues, but I think you should mention that any participants who will not complete taking part will be clearly stated in your findings.

Response: As we had listed in Table 1, the exclusion criteria include neonates receiving mechanical ventilation or continuous positive airway pressure, with skin abnormalities in the nasopharynx and/or oropharynx, with contraindication to application of skin sensors, with known arrhythmia, presence of a congenital abnormality requiring major surgical intervention, and/or any medical or psychosocial condition or circumstance that, in the opinion of the investigators, would interfere with the conduct of

the study or for which study participation might jeopardize the neonate's health. We are including all eligible infants who meet inclusion criteria and do not meet these exclusion criteria. As requested, we have included a statement clarifying that "If withdrawn by the study team, any enrolled neonate who completes at least one hour of monitoring will be included in the analysis and results."

Please can you state how long data will be stored, where it will be stored, and how it will be destroyed at the end of the storage period?

Response: As requested, we have included more information on storage of data. Of note, in the Data management section, we had included, "Data are maintained in databases hosted at the study site. All data management activities are in compliance with International Council on Harmonization (ICH) GCP E6, sponsor organization, and institutional requirements for the protection of children and confidentiality of personal and health information."

You need to be clear about dates in your proposal. How long will the study last? Your current protocol is dated June 2019 - can you please clarify if this is the enrolment date? What will your anticipated start date be?

Response: As requested, we have included the planned start and estimated study duration in the Methods section.

Please can you change use of [HCPs] for "Healthcare Providers"? This term typically refers to [Health Care Professions] in the literature, and therefore maybe misleading. I suggest "HPs" instead.

Response: As requested, we have removed the use of HCP from the manuscript.

Page 13, line 212: One of your observations will be during feeding. Will this be tube feeding, oral feeding, i.e. breast / bottle? It is important to differentiate as there are likely to be differences in instrumental readings between different methods, and also if this is one of the first oral feedings offered. Will you be using any early feeding / infant state checklists as part of your observations?

Thank - you.

Response: Observations will take place during feeding and the type of feeding will be noted.

Reviewer: 3

Reviewer Name: Dr Michelle Scoullar

Institution and Country: Burnet Institute, Australia

Please state any competing interests or state 'None declared': None declared

Please leave your comments for the authors below

see attached doc.

Response: Thank you, we have reviewed suggested edits in attached document.

Reviewer: 4

Reviewer Name: Dr. S. Lava

Institution and Country: Pediatrics, University Hospital of Lausanne, Lausanne, Switzerland

Please state any competing interests or state 'None declared': None

Please leave your comments for the authors below

Ginsburg and colleagues aim at investigating noninvasive multiparameter, continuous physiological monitoring devices in Kenya.

Their rationale starts from the recognition of the high neonatal mortality in sub-Saharan Africa. They believe that innovations in technologies that allow for early detection of major morbidities “hold great promise in helping to reduce current and projected neonatal mortality rate”. Their study rationale therefore moves from the belief that “multiparameter continuous physiological monitoring devices could be instrumental in identifying neonates at risk” and that the two noninvasive continuous physiological monitoring devices might better perform, with respect to this task, than currently available, traditional monitoring devices, since, in their opinion, current “devices are expensive and require specialized training to operate”.

The problem of high perinatal and neonatal mortality is a major worldwide problem, mainly driven by high mortality in low-income countries, like sub-Saharan Africa. Efforts to improve this situation are praiseworthy. Since a lot of such interventions are more political than medical, medical efforts should mainly aim at improving care, education and research. This study is therefore interesting.

However, I see a lot of problems with the rationale of this study:

1) Authors claim that “Innovations in neonatal care, particularly technologies that allow for early detection and intervention for major morbidities, hold great promise in helping to reduce current and projected neonatal mortality rates”. Even if this is probably true, if they want to focus on sub-Saharan Africa, several other interventions should be prioritized and hold promise of much more impact: improving hygiene, delivery at the hospital or appropriate institutions, availability of trained personnel, preterm birth prevention, train personnel to recognize clinical signs of deterioration (e.g. hypoglycemia, sepsis, respiratory distress, ...), accessibility to clean water and basic medications, ... In fact, clinical suspicion has been demonstrated to be superior to any laboratory or apparative diagnostic tool in detecting risky clinical situations.

Response: We agree with the reviewer that there are many other additional innovations and interventions that need to be prioritized and could have significant potential impact. We believe that having monitoring tools in settings where high-risk neonates are being cared for and where there are no good monitoring tools will also have significant potential lifesaving impact.

2) Authors claim that “continuous physiological monitoring is standard of care in high-resource settings” but this only apply to about 10% of newborns, who are born preterm or who need some specific care. >80% of newborns do not need any monitoring, even in high-resource countries.

Response: We have clarified this sentence to specify “for those who require it.”

3) It is unclear, while these devices should directly be tested in Africa. The potential of user-friendly manipulation and of cheaper costs also applies to high-income countries. Performing research with the fragile African population (who might be induced to participate in research by the hope of receiving better care) needs a solid rationale (either devices explicitly developed and designed for the use solely, or at least mainly, in Africa, or second-step research after having been studied in high-income countries). Were these devices developed explicitly for use in Africa? Was it verified that these populations will (be able to) buy these devices? Will they be so cheap?

Response: There is a call for more research to be done in Africa specifically in African populations. These devices are being developed and designed explicitly for use in Africa and other low-resource settings. Of note, these devices have been studied previously in high-resource settings (Israel for EarlySense and U.S. for Sibel); however, if they are to be adapted for use in low-resource settings, they need to be studied in the populations in which they will be used. Developing the market for these devices in low-resource settings is out of the scope of this present project but if successful, would be the next step.

4) Are these devices really cheaper than traditional monitors? Please provide a direct cost comparison.

Response: These investigational devices are in development and we are not yet able to provide a direct cost comparison, but the intent is that they will be less expensive than current traditional monitors.

5) “This requires development of a robust testing platform that appropriately mimics conditions common in African newborns” (lines 93-94). Sorry, but I do not agree. You might also directly test the devices in the real-world setting of everyday clinical life, without the need of complicating the research process by developing a “robust testing platform”. Furthermore, how is a robust platform defined? What does it need in order to be robust? Additionally, how it will be verified and tested that the platform is robust?

Response: For the purpose of this research, we are not evaluating the testing platform, but rather the investigational devices. Of note, as the reviewer suggests, we will be evaluating the devices in the real-world setting of everyday clinical life at Pumwani Maternity Hospital during the clinical feasibility phase as detailed in the manuscript.

6) Lines 122-123: In Africa the problem is probably mainly not to detect the alarms, but to have enough trained people to appropriately react to the alarms and the availability of the consequent therapies (e.g. non-invasive and invasive ventilation, oxygen, antibiotics and drugs, nutritional supplements, ...). Even a step before, the main problem is probably availability of health-care institutions and hospitals (with adequate electrical energy and oxygen supplies). To alarm people without the possibility of appropriately reacting to the detected situations is of no interest.

Response: We agree with the reviewer that identifying neonates at risk or experiencing a critical event is not enough without providing necessary care. Increasingly, the issue is not lack of resources, but rather poor quality of care. Appropriate monitoring can help drive improved quality of care, especially

supporting less experienced healthcare providers to provide early interventions before critical deterioration. There are numerous other efforts to improve availability of interventions and healthcare provider training.

7) Newborns can be enrolled several times (lines 140-141), which might insert several biases.

Response: We plan to acknowledge and adjust for these potential biases when analyzing and writing up the results.

8) Lines 217-219: thresholds should be clearly defined in the protocol, and not be a work-in progress process. According to the current manuscript and protocol, thresholds will be determined by an arbitrary combination of repeated observations performed on the reference device during the verification phase, international standards, and clinical expert consensus opinion. This definition is not enough clear and is therefore not really “a priori defined”.

Response: As requested, we have clarified that the setting of thresholds is not determined by an arbitrary process. The process for each variable is different but clearly defined. Repeated within and between subject data informs the most stringent threshold (it is not possible to use a threshold included within the normal individual physiological variability). For variables such as oxygen saturation for example, international performance standards are available. This is not true for most other variables. The thresholds will be defined before (a priori) any accuracy testing is undertaken. As requested, we have clarified that “Thresholds are determined largely based on repeated within and between subject observations during verification of the reference devices. This is complemented by previously published international standards where available, and clinical expert consensus opinion as needed.”

9) Your study design is “dangerous” with respect to study feasibility and solidity. In your first step, you evaluate just 2 devices. If they will not pass the first phase, your study will already stop. If you want to keep such a “2-step design”, you should include in this first phase much more devices (for example, with the aim of finally selecting the 2 best performing devices). In this version, your study entails a big risk to be abandoned already after the very few steps (but after having already included several children).

Response: The goal of this research is to evaluate these 2 specific investigational devices and if they do not pass the first accuracy phase, to stop. We do not believe this is dangerous as we would not want to proceed with evaluating a device that is not first established to be accurate. Evaluating additional devices is out of the scope of this present project.

Several further points are also critical.

- Arbitrary choice of the sample size, without any calculation

Response: As requested, we have clarified the sample size justification for each phase. “A total of up to 500 neonates are enrolled. For the verification of the reference devices at AKU-N, up to 30 neonates are enrolled. Once this initial testing and data collection of the reference devices are complete, for the accuracy evaluation phase at AKU-N, up to 120 neonates per investigational device are enrolled. Sample size estimates for the verification of the reference devices and the accuracy evaluation phase are based on the confidence intervals (CIs) desired for the limits of agreement.

Sample sizes of 100-200 typically provide tight CIs. A sample of 20 neonates with 10 replications per neonate per device per round of testing provides limits of agreement with 95% CIs ± 0.24 , calculated as $1.96 \cdot \sqrt{3/(20 \cdot 10)}$, times the standard deviation of the paired differences. The paired differences are from the reference device and manual measurements obtained during verification of the reference device, and from the reference device and investigational device measurements obtained during the accuracy evaluation phase. For the clinical feasibility phase at PMH, up to 120 neonates per investigational device are enrolled. The sample sizes for each phase have been selected to maximize the amount of information collected within the confines of the available resources.”

· The manuscript is much too long.

Response: We have tried to edit and only include what is necessary in the manuscript while also being responsive to all reviewers' requests.

· This manuscript more resembles a submission to the Ethical committee than a scientific paper: several “standard protocol sentences”, which are however general and not specific, or even do not add any relevant content to the study methods (e.g. “Standardized training, supervision, and oversight are undertaken to ensure quality, consistency, and harmonized trial procedures and implementation”: this is generally requested for every trial, but... what does the reader know, after reading this sentence, which he did not know before? Such a sentence is mandatory in a study protocol for ethical submission (and is often copy-pasted from study protocol templates), but does not add anything to a paper presenting the study methods, with the aim of being shared with the medical and scientific community...).

Response: In the Efforts towards rigorous protocol section, we have noted that “Standardized study-specific training, supervision, and oversight are undertaken to ensure quality, consistency, and harmonized trial procedures and implementation.” This is a study with many complicated measurements that have to be recorded simultaneously. We have taken great care to provide standardized study-specific training and monitoring to ensure neonate safety and accurate, high-quality data collection.

· Line 121: what do you mean with “downtime performance”? In a downtime period, a device is supposed not to work... how can its performance during a “break” time be evaluated? Why should this be relevant?

Response: All monitoring devices, even in high-resource settings, have periods of time during which they do not display valid clinical data. This is typically caused by dislodgement of sensors or other external interference. This downtime depends on the type of sensor and the clinical context.

· Line 124: “delay time”: is the device so slow in transmitting information to have a measurable delay? Should the transmission not be instantaneous (the delay between clinical event, detection by the device and transmission to the central being $<1s$)?

Response: All clinical alarms have some delay. One of the most important ways to reduce false alarms is to introduce a delay that is clinically acceptable. There will always be a trade-off between false alarms and delays. This is not a delay in electronic communication.

· Lines 127-128: what is the difference between “verification” and “accuracy evaluation”? Please be as explicit and detailed as possible.

Response: As we had noted in the Study procedures section, the verification is for the reference devices only and is undertaken prior to initiating the accuracy evaluation of the investigational devices to ensure the reference devices are robustly functional and to confirm the reference devices within subject repeatability and accuracy compare to standard clinical measurements for the relevant parameters of interest. The reference device measurements will be compared to manual measurements, clinical monitor observations, and video-assisted observations. The accuracy evaluation is to compare the investigational devices to the reference devices.

· Line 254: “qualitative”: you mean “quantitative”, well?

Response: No, we mean qualitative. There are separate consent forms for the quantitative and qualitative components. However, we can omit the word “qualitative” to avoid confusion.

· Line 256: “observations [...] by trained qualitative staff”: how will they be trained? What will they assess and how? This description is too "general" and not enough detailed and objective. Please provide a list of items/data they will address, observe and collect.

Response: As requested, we have added details to clarify the qualitative component of the evaluation.

· Lines 335-336: “An a priori–defined accuracy margin for agreement is used”: please transparently state this a priori defined margin (with its minimum and maximum) in the protocol.

Response: As requested, we have clarified that determination of the accuracy margin requires the data obtained from the verification of the reference devices to determine the within and between subject variability.

· Lines 341-342: “sensitivity, specificity, positive predictive values and negative predictive values”: for which events? How will these events be defined and decided? (e.g. a possible event might be "desaturation". This should be defined as "desaturation at <...%", with rationale for the choice of that specific value).

Response: As requested, we have clarified that threshold values will be defined for each device depending on their specific defined operating performances. We will set high and low threshold values for heart rate, respiratory rate, oxygen saturation (only if present), and apnea detection. Thresholds will be selected by the coinvestigators. The same thresholds will be used for both investigational and reference devices.

· Lines 347-348: which questions will be asked in these semi-structured questionnaires? In a paper, which uses 20 pages to describe the study methods, this information must be provided.

Response: As requested, we have added details to clarify the qualitative component of the evaluation.

· Line 365: why approval (also) from USA, if the study is performed in Kenya?

Response: We typically get an external ethics review in the U.S. for all of our clinical research studies.

· Line 381: “ANNE sensor system and neonates will be closely”  “ANNE sensor system, if this device will pass the validation phase, and neonates will be closely”

Response: There is no validation phase. There is a verification but this is for the reference devices only. The ANNE sensor system will be evaluated for accuracy and whenever it is applied to an enrolled neonate, the neonate will be closely monitored for any adverse events.

· Lines 401-403: are these really limitations? I feel, they are rather strengths: a good device should be reliable in a wide range of "timepoints of day", clinical conditions, gestational age, weight range, stability degrees, ...

Response: We have clarified this section to reflect that they may also be seen as strengths.

· Line 407: again, this is not a limitation. You are not evaluating a therapy, but the reliability of a diagnostic device: it should be reliable across a large spectrum of situations. If the test is standardized, the results are not generalizable and, therefore, not extrapolable to everyday clinical life.

Response: We have clarified this section to reflect that this may also be seen as a strength.

VERSION 2 – REVIEW

REVIEWER	Dr Michelle Scoullar Burnet Institute, Australia
REVIEW RETURNED	03-Jan-2020

GENERAL COMMENTS	· Regarding both devices: o What environmental conditions can these devices operate in? specifically temperature and humidity limits of operation? What fluctuations can they tolerate in these parameters, e.g for shipping and storage of equipment. Response: Including all the device specifications that would be included in the FDA submission and this level of detail would make this manuscript quite long. Given that another reviewer also commented that “the manuscript is much too long,” we have tried to be concise and only include what is necessary in the manuscript while also being responsive to all reviewers’ requests. Feb2020: I agree, you do not need to include every specification, but it would be worth reassuring readers that you have closely considered the
---

	core requirements and are satisfied that these are met. Or at the very least referring to a paper / document / manufacturing specifications that detail that these devices are known to tolerate the required temperature and humidity they are likely to encounter.
--	--

REVIEWER	Dr. S. Lava Pediatrics, University Hospital of Lausanne, Lausanne, Switzerland
REVIEW RETURNED	02-Jan-2020

GENERAL COMMENTS	The aim of improving healthcare of African children is praiseworthy. However, the Authors failed to address several (most) of my previous comments. Unfortunately, they did not take the opportunity to improve their study design and address some critical points. In my opinion, this study protocol still contains several weaknesses and this manuscript is not of sufficient scientific interest to be shared in an internationally recognised peer-reviewed journal. Furthermore, even simple, “not-design-related”, presentation comments were ignored (here only 2 examples: 1) I requested to shorten the manuscript, but the revised manuscript is 256 words longer than the originally submitted version; 2) I requested to list some points as strengths instead of as weaknesses, but the Authors did not revise their sentence and simply added “These limitations may also be viewed as strengths.”). Finally and noteworthy, I do not understand the interest to submit the study protocol to a scientific journal and to a reviewers’ panel, if the study (as it can be read in the revised version of the manuscript) already started: obviously, the comments and proposals for design improvement cannot be intergated any more in the study design.
---

VERSION 2 – AUTHOR RESPONSE

Reviewer(s)' Comments to Author:

Reviewer: 4

Reviewer Name

Dr. S. Lava

Institution and Country

Pediatrics, University Hospital of Lausanne,
Lausanne, Switzerland

Please state any competing interests or state ‘None declared’:
I do not have any conflict of interest with respect to this study protocol.

Please leave your comments for the authors below

The aim of improving healthcare of African children is praiseworthy. However, the Authors failed to address several (most) of my previous comments. Unfortunately, they did not take the opportunity to improve their study design and address some critical points. In my opinion, this study protocol still contains several weaknesses and this manuscript is not of sufficient

scientific interest to be shared in an internationally recognised peer-reviewed journal. Furthermore, even simple, “not-design-related”, presentation comments were ignored (here only 2 examples: 1) I requested to shorten the manuscript, but the revised manuscript is 256 words longer than the originally submitted version; 2) I requested to list some points as strengths instead of as weaknesses, but the Authors did not revise their sentence and simply added “These limitations may also be viewed as strengths.”). Finally and noteworthy, I do not understand the interest to submit the study protocol to a scientific journal and to a reviewers’ panel, if the study (as it can be read in the revised version of the manuscript) already started: obviously, the comments and proposals for design improvement cannot be intergated any more in the study design.

Response: We remain grateful for Reviewer 4’s review and comments. We did our very best to address and incorporate as appropriate all of Reviewer 4’s comments and points for clarification (as well as those of the three other reviewers), while also being mindful of the length. After careful consideration, we do not think it is necessary to change the study design. We feel we have adequately addressed all of Reviewer 4’s comments.

Reviewer: 3

Reviewer Name

Dr Michelle Scoullar

Institution and Country

Burnet Institute, Australia

Please state any competing interests or state ‘None declared’:
none declared

Please leave your comments for the authors below

- Regarding both devices:
 - o What environmental conditions can these devices operate in? specifically temperature and humidity limits of operation? What fluctuations can they tolerate in these parameters, e.g for shipping and storage of equipment.

Response: Including all the device specifications that would be included in the FDA submission and this level of detail would make this manuscript quite long. Given that another reviewer also commented that “the manuscript is much too long,” we have tried to be concise and only include what is necessary in the manuscript while also being responsive to all reviewers’ requests.

Feb2020:

I agree, you do not need to include every specification, but it would be worth reassuring readers that you have closely considered the core requirements and are satisfied that these are met. Or at the very least referring to a paper / document / manufacturing specifications that detail that these devices are known to tolerate the required temperature and humidity they are likely to encounter.

Response: As requested, we have added references 6 and 7 to support the study of these devices in this setting under expected environmental conditions.